# PDEformer: Towards a Foundation Model for One-Dimensional Partial Differential Equations

**Zhanhong Ye[1], Xiang Huang[2], Leheng Chen[1], Hongsheng Liu[2], Zidong Wang[2], Bin Dong[1,3]***

[1]Beijing International Center for Mathematical Research, Peking University, Beijing, China
[2]Central Software Institute, Huawei Technologies Co. Ltd, Hangzhou, China
[3]Center for Machine Learning Research, Peking University, Beijing, China
`{yezhanhong,chenlh}@pku.edu.cn`
`{huangxiang42,liuhongsheng4,wang1}@huawei.com`
`dongbin@math.pku.edu.cn`

## Abstract

This paper introduces PDEformer, a neural solver for partial differential equations (PDEs) capable of simultaneously addressing various types of PDEs. We propose to represent the PDE in the form of a computational graph, facilitating the seamless integration of both symbolic and numerical information inherent in a PDE. A graph Transformer and an implicit neural representation (INR) are employed to generate mesh-free predicted solutions. Following pretraining on data exhibiting a certain level of diversity, our model achieves zero-shot accuracies on benchmark datasets that is comparable to those of specifically trained expert models. Additionally, PDEformer demonstrates promising results in the inverse problem of PDE coefficient recovery.

## 1 Introduction and Related Work

The efficient solution of PDEs plays a crucial role in various scientific and engineering domains, from simulating physical phenomena to optimizing complex systems. In recent years, many learning-based PDE solvers have emerged. Some methods (Raissi et al., 2019; Sirignano & Spiliopoulos, 2018; Ee & Yu, 2017; Zang et al., 2020) represent the approximate PDE solution with a neural network, and are tailored to individual PDEs. Other approaches, such as neural operators like Fourier Neural Operator (FNO) (Li et al., 2021) and DeepONet (Lu et al., 2021), tackle parametric PDEs by taking the PDE parameters (coefficient fields, initial conditions, etc.) as network inputs. While these methods exhibit a higher level of generality, their capability is still limited to solving a specific type of PDE.

Drawing inspirations from successful experiences in natural language processing and computer vision, we aim to develop a foundation PDE model with the highest generality, capable of handling any PDE in the ideal case. Given a new PDE to be solved, we only need to make a direct (zero-shot) inference using this model, or fine-tune it only for a few steps using a relatively small number of solution snapshots. By leveraging the power of generality, foundation models have demonstrated great potential in capturing the similarity inherent in a wide range of tasks, and producing high-quality feature representations that are beneficial to various applications (Bommasani et al., 2022; Zhou et al., 2023). Specific to the realm of scientific computing, we anticipate such a foundation PDE model can achieve high solution accuracy, which is comparable with or even surpass expert models that are trained to solve a specific type of PDE. Besides, it should be easily adapted to tackle with down-stream tasks, including inverse problems, inverse design, optimal control, etc.

A PDE to be solved would involve two parts of information: one is the symbolic part specifying the mathematical form of the PDE, and the other is the numeric part that includes the PDE coefficients, initial and boundary values, etc. Typical neural operators like FNO and DeepONet deal with a

---

*Correspondence to `dongbin@math.pku.edu.cn`.

specific form of PDE, and only need to take the numeric information as the network input. However, in order to construct a foundation model generalizable to different PDEs, the symbolic information has to be integrated seamlessly.

Some existing approaches towards this direction (Lorsung et al., 2023; Yang et al., 2024; Liu et al., 2023) employ a language model, where the mathematical expression of the PDE serves as the input. These methods may struggle to fully capture the complex interaction between the symbolic and the numeric information. Other strategies avoid explicit input of the PDE forms, opting to encode it implicitly in the numeric input to the model. For example, Yang et al. (2023) and Yang & Osher (2024) use several parameter-solution pairs of the target PDE. Specific to time-dependent PDEs, McCabe et al. (2023) trains a model to predict the next time-step based on a few history solution snapshots, which contain the information of what the underlying dynamics is. Subramanian et al. (2023) specifies the PDE to be solved by the location of the nonzero input channels. Such implicit input methods could be insufficient for encoding classes of PDEs with greater variaty and complexity, and may have to be accompanied with another solver to prepare the additional solution snapshots.

In this paper, we introduce PDEformer. Different from previous approaches, we propose to express the symbolic form of the PDE as a computational graph, ensuring that the resulting graph structure, along with its node types and feature vectors, encapsulate all the symbolic and numeric information necessary for solving the PDE. A graph Transformer and an INR are utilized to generate mesh-free predicted solutions. After pretraining on PDEs with a certain level of diversity, evaluation on benchmark datasets shows that PDEformer exhibits higher zero-shot prediction accuracy compared with baseline expert models, or can achieve this after fine-tuning with limited data. The potential of application to various down-stream tasks is primarily validated by the PDE coefficient recovery inverse problem. Although our experiments are currently limited to one-dimensional PDEs, we believe it would serve as a noteworthy milestone towards building a foundation PDE model.

## 2 METHODOLOGY

We consider 1D time-dependent PDEs on $(t, x) \in [0, 1] \times [-1, 1]$ with periodic boundary conditions, of the general form

$$\mathcal{F}(u, c_1, c_2, \dots) = 0, \quad u(0, x) = g(x),$$

where $c_1, c_2, \dots \in \mathbb{R}$ are real-valued coefficients, and $g(x)$ is the initial condition. Here, we assume the operator $\mathcal{F}$ has a symbolic expression, which may involve differential and algebraic operations. The goal is to construct a surrogate of the solution mapping $(\mathcal{F}, g, c_1, c_2, \dots) \mapsto u$ that essentially takes the form of the operator $\mathcal{F}$ as its input. We illustrate the overall network architecture in Figure 1. A primary intepretation of the elements involved will be presented in the following text, with further details left for the appendix.

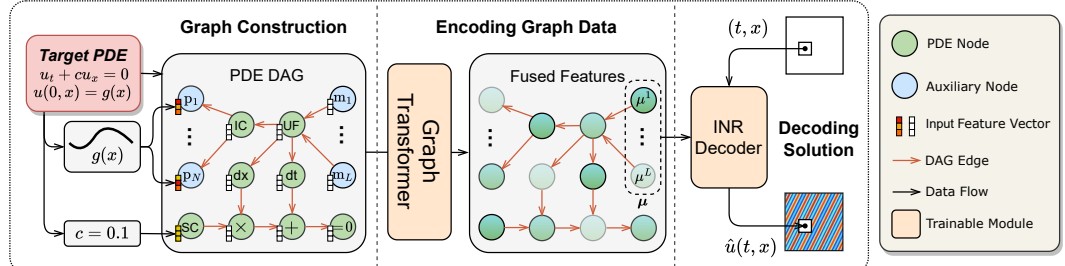

Figure 1: PDEformer architecture, taking $\mathcal{F}(u, c) = u_t + cu_x$ as the example.

**Graph Construction** We first represent $\mathcal{F}$, i.e. the symbolic information specifying the PDE form, as a computational graph. In such a computational graph, a node may stand for an unknown field variable (denoted as UF), a scalar coefficient (SC), the initial condition (IC), as well as a differential or algebraic operation, and a directed edge can be used to specify the operands involved in an operation. This would constitute of a directed acyclic graph (DAG) with heterogeneous nodes and homogeneous edges.

Then, in order to include the numeric information, we endow each graph node with a feature vector in $\mathbb{R}^{d_f}$. For a scalar coefficient $c$, the value is repeated $d_f$ times to form the feature vector of the

corresponding SC node. In terms of the initial condition $g(x)$, we assume it is given at an equi-spaced grid with $n_x$ points. Inspired by ViT (Dosovitskiy et al., 2021), we divide these grid values into $N = n_x/d_f$ patches, yielding $N$ vectors $\boldsymbol{g}_1, \ldots, \boldsymbol{g}_N \in \mathbb{R}^{d_f}$. These will be used as the feature vectors of the $N$ newly-introduced "patch" nodes, whose types are denoted as $\mathrm{p}_1, \mathrm{p}_2, \ldots, \mathrm{p}_N$, respectively. We shall connect these patch nodes with the corresponding IC node. The feature vectors of all the remaining nodes are set as zero.

Moreover, we introduce $L$ additional nodes with type $\mathrm{m}_1, \mathrm{m}_2, \ldots, \mathrm{m}_L$, and connect them to the corresponding UF node. These nodes will be used to decode the predicted solution as explained below.

**Encoding Graph Data**  The symbolic and numeric information encapsulated in the graph data is integrated into a latent code $\boldsymbol{\mu} = [\mu^1, \ldots, \mu^L]^{\mathrm{T}} \in \mathbb{R}^{L \times d_e}$. This is accomplished by the graph Transformer, a class of modern Transformer-based graph neural networks with impressive representational capabilities. An adapted version of Graphormer (Ying et al., 2021) is utilized as the specific architecture in the experiments, while more potential alternatives can be found in Min et al. (2022). For $\ell = 1, \ldots, L$, we let $\mu^\ell \in \mathbb{R}^{d_e}$ be the embedding vector assigned to the node with type $\mathrm{m}_\ell$ in the output layer of this graph Transformer.

**Decoding the PDE Solution**  We employ an INR that takes the coordinate $(t, x)$ as input, and produces the mesh-free prediction $\hat{u}(t, x)$ according to $\boldsymbol{\mu}$. Various INR architectures with such an external condition have been adopted in neural operators (Yin et al., 2023), data compression (Dupont et al., 2022) and generative models (Singh et al., 2023). In the experiments, we utilize an adapted version of Poly-INR (Singh et al., 2023) with $L$ hidden layers due to its efficiency, and the modulations of the $\ell$-th hidden layer is generated based on $\mu^\ell$.

## 3  RESULTS

### 3.1  PRETRAINING STAGE

We generate a dataset containing 500k samples, distinguished by equation types, coefficients and initial conditions. Specifically, the addressed PDEs follow the form[1] $u_t + f_0(u) + f_1(u)_x - \nu u_{xx} = 0$, $(t, x) \in [0, 1] \times [-1, 1]$ with periodic boundaries and initial condition $u(0, x) = g(x)$, $x \in [-1, 1]$, where $f_i(u) = \sum_{k=0}^{3} c_{ik} u^k$ for $i = 0, 1$. The corresponding PDEs are solved using randomly generated $c_{ik}, \nu$, and $g(x)$ with the Dedalus package (Burns et al., 2020) to create the pretraining dataset. Pretraining involved 1,000 epochs on 90% of the data, reserving the remaining 10% for testing. PDEformer achieves a relative $L^2$ error of 0.0104 on the training dataset, and 0.0128 on the test dataset. Figure 2 illustrates the pretrained PDEformer's predictions on the test dataset, emphasizing its high accuracy and proficiency in learning representations across diverse PDEs.

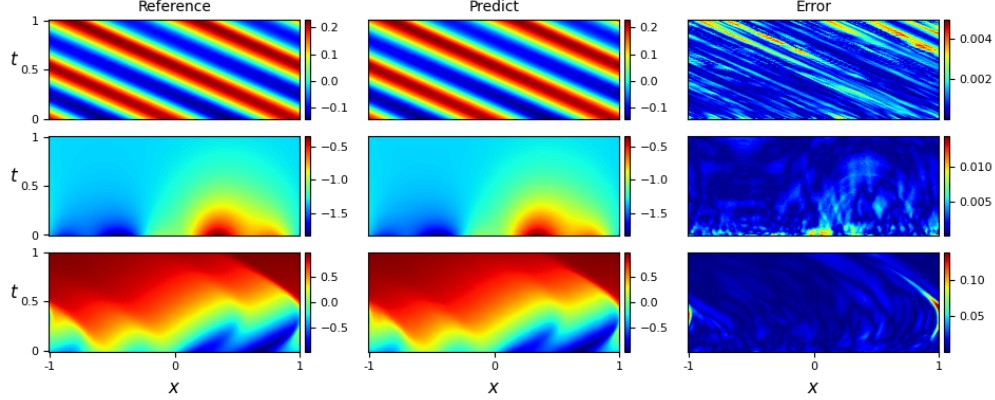

Figure 2: Comparison of prediction results obtained from the pretrained PDEformer on the test dataset with reference solutions. Each row in the figure represents a single sample, and these three samples were randomly selected.

---

[1]Terms with zero coefficients are removed, and not taken as the input of PDEformer. The PDEs involved have different equation types in this sense.

## 3.2 FORWARD PROBLEM

The pretrained PDEformer is highly versatile in handling various equations. Its performance on forward problems is evaluated using parametric PDEs from PDEBench (Takamoto et al., 2022), including Burgers', Advection, and 1D Reaction-Diffusion PDEs. Comparative analysis is conducted with neural operator models tailored for individual PDEs. In this context, neural operators receive initial conditions and predict the entire solution field. Notably, all previous methods as well as `PDEformer-FS` are trained from-scratch and tested separately on different datasets, while the `PDEformer` model shows zero-shot inference across all test datasets post pretraining. Furthermore, the `PDEformer-FT` model involves an additional fine-tuning process on the corresponding PDEBench dataset.

In Table 1, the pretrained PDEformer model showcases zero-shot proficiency by attaining reasonable accuracy in all in-distribution tests[2]. Remarkably, for Burgers' equation with $\nu = 0.1$ and $0.01$, the zero-shot PDEformer outperforms all the baseline models trained specifically on these datasets. Such superior performance can be partially attributed to the network architecture we have utilized, as PDEformer already exhibits competitive performance when trained from-scratch. We believe that the additional improvement of the pretrained PDEformer stems from its exposure to diverse PDEs during pretraining, from which the model may learn a generalizable law to outperform models trained specifically for individual PDEs. Results of the out-of-distribution tests can be found in Table 3 in the Appendix. The fine-tuned PDEformer consistently excels in all in-distribution and out-of-distribution tests, further highlighting the robustness and versatility of our approach in solving a wide range of PDEs.

Table 1: Test relative $L^2$ error on PDEBench, in which the PDE coefficients lie within the range of the pretraining data. We format the first and second best outcomes in bold and underline, respectively.

| Model | Burgers | | Advection |
|---|---|---|---|
| | $\nu = 0.1$ | $\nu = 0.01$ | $\beta = 0.1$ |
| U-Net (Ronneberger et al., 2015) | 0.1627 | 0.2253 | 0.0873 |
| Autoregressive U-Net | 0.2164 | 0.2688 | 0.0631 |
| DeepONet (Lu et al., 2021) | 0.0699 | 0.1791 | 0.0186 |
| FNO (Li et al., 2021) | 0.0155 | 0.0445 | 0.0089 |
| PDEformer-FS (Ours) | 0.0135 | 0.0399 | 0.0124 |
| PDEformer (Ours) | 0.0103 | 0.0309 | 0.0119 |
| PDEformer-FT (Ours) | **0.0046** | **0.0146** | **0.0043** |

Thanks to the high-quality initialization obtained after pretraining, general foundation models are known to exhibit efficient adaptation to new tasks (Bommasani et al., 2022; Zhou et al., 2023). To compare the efficiency of fine-tuning PDEformer with training traditional expert models, we conduct a comparative analysis on the Advection equation ($\beta = 1$, OoD) dataset with a limited number of 100 training samples. As depicted in Figure 3, PDEformer rapidly reaches convergence in about just 100 iterations. Conversely, the FNO model, trained from scratch, results in a higher test error even after several thousands of iterations. Indeed, it is possible for traditional neural operators to start from a better initialization. However, designed for a specific type of PDE, they cannot be pretrained on 500k data samples containing diverse PDEs as PDEformer does. The valid option left for us is to pretrain them on one different PDE, and then transfer to the target setting, which could be much less efficient. Post pretraining on 9k samples of the Advection equation with $\beta = 0.1$ for 1k iterations, the `FNO-FT` model only exhibits a limited improvement over the corresponding from-scratch version, as can be seen in the figure. This contrast highlights the pretrained PDEformer's swift and accurate adaptability, marking a significant advancement over existing expert models.

---

[2]Here, "in-distribution" and "out-of-distribution"(OoD) refer to the range of the PDE coefficients. In-distribution PDEs have coefficients lying within the range of the pretraining data, whereas OoD samples fall outside this range. Note that PDEBench datasets labeled as "in-distribution" are not utilized during pretraining. Therefore, we term the corresponding PDEformer inference as "zero-shot".

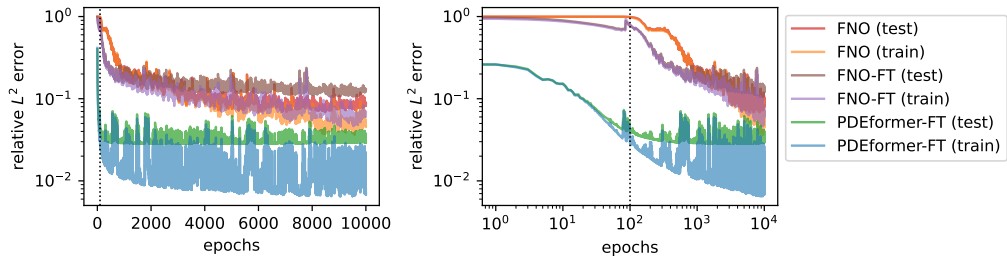

Figure 3: Comparing the speed of fine-tuning PDEformer with training FNO from scratch and fine-tuning a pretrained FNO. The right subfigure uses a logarithmic scale for the $x$-axis, whereas the left employs a linear scale. The vertical lines correspond to 100 iterations.

### 3.3 INVERSE PROBLEM

In addition to forward problems, we can leverage the pretrained PDEformer to the PDE coefficient recovery inverse problem based on one noisy observed solution instance. For each PDE, we feed the current estimation of PDE coefficients into the pretrained PDEformer to get the predicted solutions, and minimize the relative $L^2$ error against the observations to obtain the recovered coefficients. As this optimization problem exhibits a lot of local minima, the particle swarm optimization algorithm (Wang et al., 2018) is utilized. Figure 4 illustrates the outcomes involving 40 PDEs from the test set explained in Section 3.1. The number of coefficients to be recovered varies for each equation (ranging from 1 to 7). In the absence of noise, the recovered coefficients closely align with the ground-truth values, with scattered points primarily distributed along the $y = x$ line. The existence of a few outliers could be attributed to the intrinsic ill-posed nature of this inverse problem. Even under high noise levels, the majority of the PDE coefficients can be effectively recovered.

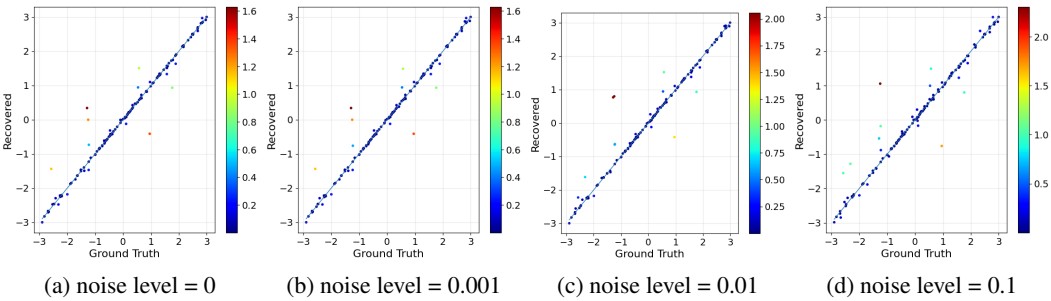

(a) noise level = 0    (b) noise level = 0.001    (c) noise level = 0.01    (d) noise level = 0.1

Figure 4: Results of the PDE coefficient recovery problem under various noise levels. For every PDE, all non-zero coefficients are recovered, with each coefficient depicted as a point in the figure. Consequently, the number of points displayed exceeds the number of PDEs involved.

ACKNOWLEDGMENTS

This work is supported in part by the National Science and Technology Major Project (2022ZD0117804). Bin Dong is supported in part by the New Cornerstone Investigator Program.

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

APPENDIX

In the Appendix, we offer comprehensive supplementary materials to enhance the understanding of our study and support the reproducibility of our results. Appendix A delves into the details of our computational graph representation, elucidating its design and the rationale behind its structure. In Appendix B.1, we present an overview of the datasets employed during the pretraining stage of our models, including data sources and preprocessing steps. Appendix B.2 explains the PDEBench datasets' features and our postprocessing steps. Appendix C.1 explores the underlying architecture of our Graph Transformer, detailing its components and their difference with original Graphormer. In Appendix C.2, we present the detailed architecture of the Poly-INR with hypernets. Appendix C.3 outlines the specific training parameters and settings, offering clarity on the experimental setup and execution. Appendix D provides more detailed results of the experiments. Finally, Appendix E compares the inference time of different neural network models and the traditional solver.

## A    DETAILED INTEPRETATION OF THE COMPUTATIONAL GRAPH REPRESENTATION

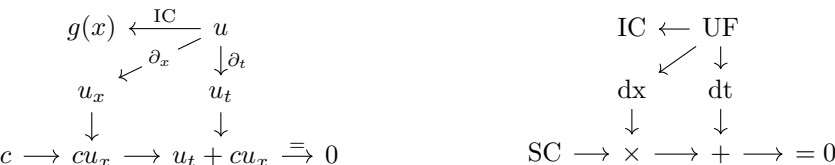

Figure 5: Illustration of how the form of a PDE can be represented as a computational graph, taking the advection equation $u_t + cu_x = 0, u(0, x) = g(x)$ as the example. The left panel shows the logical meaning of the nodes and edges, and the right panel illustrates the formalized data structure that is taken as the input of PDEformer. We also note that, different from textual representations, this formalization of DAG is independent of the choice of symbols and the order of addition or multiplication. For example, the equation $\beta v_x + v_t = 0, v|_{t=0} = v_0$ also corresponds to the DAG shown on the right panel.

Figure 5 gives an illustration of the semantic meanings of the computational graph. We also have the following remarks on the computational graph representation of the PDE form:

- Only a small number of node types are involved in this computational graph: `UF` (unknown field variable), `SC` (scalar coefficient), `IC` (initial condition), $\mathrm{dt}, \mathrm{dx}$ (differentiation with respect to $t$ and $x$, respectively), $+$ (sum), $\times$ (product), $-$ (negation), $(\cdot)^2$ (square), $= 0$ (being equal to zero).

- Note that "$-$" stands for negation rather than subtraction in the computational graph, making it a unary rather than a binary operation. This is because nodes in this computational graph are not ordered, and the edges are homogeneous. If the binary subtraction operation is involved in the computational graph, we cannot differentiate the subtrahend and the minuend.

- For the same reasons, we do not include a power operation node. However, we note that powers with a positive interger exponent can be expressed. For example, $u^3 = u \times u^2$, and $u^{11} = ((u^2)^2)^2 \times u^2 \times u$ since $11 = 2^3 + 2^1 + 2^0$.

- Although not involved in our experiments, node types representing special functions such as $\sin, \cos, \exp, \log$ can be introduced as well, depending on the form of the PDEs involved. This also enables expression of the general power operation, since we have $a^b = \exp(b \times \log(a))$.

- Disregarding the auxiliary nodes, nodes with type `UF` and `SC` would have a zero in-degree, $+$ and $\times$ have an in-degree that is equal to or greater than two, and the in-degrees of all the remaining nodes would be exactly one.

- In terms of the auxiliary nodes, we let each patch node $\mathrm{p}_i$ receive an edge from the corresponding `IC` node, and each latent modulation node $\mathrm{m}_\ell$ emanate an edge towards `UF`. We

adopt such convention of edge direction in order to improve the connectivity of the final DAG, since we shall mask out the attention between disconnected node pairs in the graph Transformer module (see Appendix C.1).

In the experiments, the initial condition $g(x)$ is discretized on a spatial grid with $n_x = 256$ points, and we divide the values into $N = 16$ patches of length $d_f = 16$. This exhibits better solution accuracy compared with the case $N = 4$ or $N = 1$. For the external condition of the INR decoder, we observe that using different latent vectors $\mu^1, \ldots, \mu^L \in \mathbb{R}^{d_e}$ for each hidden layer leads to improved performance compared with the case of using a shared one $\mu \in \mathbb{R}^{d_e}$. The introduction of the auxiliary nodes is therefore deemed meaningful.

# B  DATASETS

## B.1  PRETRAINING DATASETS

The dataset constitutes of solutions to PDEs of the form

$$u_t + f_0(u) + f_1(u)_x - \nu u_{xx} = 0, \quad (t, x) \in [0, 1] \times [-1, 1],$$
$$u(0, x) = g(x), \quad x \in [-1, 1],$$

where $f_i(u) = \sum_{k=0}^{3} c_{ik} u^k$ for $i = 0, 1$. Each coefficient $c_{ik}$ is set to zero with probability 0.5, and drawn randomly from $U([-3, 3])$ otherwise.[3] The viscosity $\nu$ satisfies $\log \nu \sim U([\log 10^{-3}, \log 1])$. For the case of a linear flux, i.e. when $c_{12} = c_{13} = 0$, we set $\nu = 0$ with probability 0.5. Note that terms with a zero coefficient will be excluded in the computational graph of the PDE. The random initial condition $g(x)$ is generated in the same way as the PDEBench dataset, as will be explained in Appendix B.2.

The numerical solutions are obtained using the open-source Python package named Dedalus v3 (Burns et al., 2020), which is a flexible solver based on spectral methods. To generate the data samples, we use a uniform spatial grid with 256 grid points. The solver proceeds at a time-step of $\delta t_{\text{solver}} = 4 \times 10^{-4}$, and the solution snapshots are recorded with time-step $\delta t_{\text{data}} = 0.01$, yielding a total of 101 temporal values for each data sample. When Dedalus fails to solve the PDE, or when the $L^\infty$-norm of the solution exceeds 10, the corresponding data sample will be discarded, and not included in the final dataset.

As the PDEs have a periodic boundary condition, and are discretized on uniform grid points, we introduce data augmentation by a random translation along the $x$-axis during the pretraining stage. For each data instance, a total of 8192 spatial-temporal coordinate points are randomly sampled from the $101 \times 256$ grid. These sampled points are taken as the input of the solution decoder INR, and we compare the model predictions with the ground-truth numerical values to compute the loss.

## B.2  PDEBENCH DATASETS

In this subsection, we present an overview of three 1D PDE datasets derived from PDEBench, which we employed in our experimental analysis. Each dataset, tailored to a specific PDE type and coefficient configuration, encompasses 10k instances. For our training purposes, we utilized 9k samples from each dataset, reserving the remaining 1k samples for testing. It is crucial to note that all these PDEBench datasets adhere to periodic boundary conditions.

- Burgers' equation[4]: $\partial_t u + \partial_x(u^2) = \frac{\nu}{\pi} \partial_{xx} u$ for $(t, x) \in [0, 2] \times [-1, 1]$, where $\nu \in \{0.1, 0.01, 0.001\}$. This equation is a fundamental partial differential equation from fluid mechanics.

- Advection equation: $\partial_t u + \beta \partial_x u = 0$ for $(t, x) \in [0, 2] \times [0, 1]$, where $\beta \in \{0.1, 1\}$. The equation models the transport of a quantity $u$ without alteration in its form.

---

[3]The value of $c_{10}$ has no effect on the PDE solutions, and PDEformer can learn such redundancy during training. We exclude this term in the inverse problems.

[4]The convection term is $\partial_x(u^2)$ rather than $\partial_x(u^2/2)$ due to an implementation issue in the PDEBench data generation code. See https://github.com/pdebench/PDEBench/issues/51 for more details.

- Reaction-Diffusion equation: $\partial_t u = \nu \partial_{xx} u + \rho u(1-u)$ for $(t,x) \in [0,1] \times [0,1]$, where we only consider $\nu = 1, \rho = 1$. This equation represents a process combining chemical reaction and diffusion dynamics.

The initial conditions for each dataset are given by $u_0(x) = \sum_{k_i=k_1,\dots,k_N} A_i \sin(k_i x + \phi_i)$, with frequency numbers $k_i = \frac{2\pi n_i}{L_x}$, where $n_i$ are integers randomly selected within a pre-determined range and $L_x$ is the length of the spatial domain, amplitudes $A_i$ are random numbers within $[0,1]$, and phases $\phi_i$ are chosen randomly from the interval $(0, 2\pi)$. The absolute value function with a random signature, as well as restriction to a random sub-interval by multiplying a window function, are applied afterwards with $10\%$ probability each. For the Reaction-Diffusion equation, the range of the initial condition is rescaled to the unit interval $[0,1]$.

In order to utilize the pretrained PDEformer model to make predictions, we rescale the spatial-temporal coordinates to the range $(t', x') \in [0,1] \times [-1,1]$, and the resulting PDEs taken as the input of PDEformer have the following form:

- Burgers' equation: $\partial_{t'} u + \partial_{x'}(2u^2) - \frac{2\nu}{\pi} \partial_{x'x'} u = 0$, where $t' = t/2, x' = x$.
- Advection equation: $\partial_{t'} u + \partial_{x'}(4\beta u) = 0$, where $t' = t/2, x' = 2x - 1$.
- Reaction-Diffusion equation: $\partial_{t'} u - 4\nu \partial_{x'x'} u + (-\rho)u + \rho u^2 = 0$, where $t' = t, x' = 2x - 1$.

To ensure equitable comparisons among the baseline models, we standardize the resolution of all PDEBench samples to $256 \times 256$. More specifically, the original PDEBench datasets have a spatial resolution of $1024$, which is downsampled to $256$. The original number of recorded time-steps is $201$ for the Burgers and Advection datasets and $101$ for the one-dimensional Reaction-Diffusion dataset, and a linear interpolation is utilized to obtain a temporal resolution of $256$. It is important to note that PDEformer makes mesh-free predictions, enabling us to set the temporal resolution to $101$ for the pretraining dataset, and $256$ for the PDEBench dataset. For FNO and U-Net (non-autoregressive case), the initial value is repeated $256$ times to form the two-dimensional data with resolution $256 \times 256$, and then taken as the network input.

## C  NETWORK ARCHITECTURE AND TRAINING SETTING

### C.1  GRAPH TRANSFORMER ARCHITECTURE

The specific graph Transformer architecture employed in our experiments is based on Graphormer (Ying et al., 2021), with some adaptations to fit our setting. The details are presented as below.

**Initial Embedding Vector**   In the graph Transformer, the initial embedding vector of node $i$ is given as
$$h_i^{(0)} = x_{\text{type}(i)} + \text{Feat-Enc}(f_i) + z_{\text{deg}^-(i)}^- + z_{\text{deg}^+(i)}^+,$$
where $x, z^-, z^+ \in \mathbb{R}^{d_e}$ are learnable embedding vectors specified by the node type $\text{type}(i)$, indegree $\text{deg}^-(i)$ and outdegree $\text{deg}^+(i)$, respectively. In order to encode the node feature vector $f_i \in \mathbb{R}^{16}$ that is not involved in the original Graphormer (Ying et al., 2021), we utilize a feature-encoder Feat-Enc, which is a three-layer multi-layer perceptron (MLP) with ReLU activations and $256$ neurons in each hidden layer.

**Attention Bias**   Denote $\phi(i,j)$ to be the shortest path length from node $i$ to node $j$. If such a path does not exist, or has a length greater than $14$, we shall set $\phi(i,j) = 14$. For each attention head involved in the graph Transformer, the attention bias corresponding to the node pair $(i,j)$ is given as
$$B_{ij} = b_{\phi(i,j)}^+ + b_{\phi(j,i)}^- + d_{ij}. \tag{1}$$
Here, $b_{\phi(i,j)}^+$ and $b_{\phi(j,i)}^-$ are learnable scalars indexed by $\phi(i,j)$ and $\phi(j,i)$ respectively, and shared across all layers. The additional term $d_{ij}$, which does not appear in the original Graphormer, is introduced to mask out attention between disconnected node pairs. More specifically, when node $i$

and node $j$ are connected in the graph, i.e. there exists a path either from $i$ to $j$ or from $j$ to $i$, we take $d_{ij} = 0$, and set $d_{ij} = -\infty$ otherwise. We observe in our experiments that the overall prediction accuracy can be improved with such an additional masking operation. Moreover, since our graph has homogeneous edges, we do not introduce the edge encoding term that appears in the original Graphormer.

**Graph Transformer Layer**    The structure of the graph Transformer layer is the same as the original Graphormer, and we include it here for convenience to the readers. Each layer takes the form

$$\bar{h}^{(l)} = \text{Attn}(\text{LN}(h^{(l-1)})) + h^{(l-1)}$$
$$h^{(l)} = \text{FFN}(\text{LN}(\bar{h}^{(l)})) + \bar{h}^{(l)},$$

where FFN represents a position-wise feed-forward network with a single hidden layer and GeLU activation function, and LN stands for layer normalization. In terms of the self-attention block Attn, we shall follow the convention in the original Graphormer paper, and only present the single-head case for simplicity. Let $H = [h'_1, \cdots, h'_n]^\text{T} \in \mathbb{R}^{n \times d_e}$ denote the input of the self-attention module involving $n$ graph nodes, the self-attention is computed as

$$Q = HW_Q, \quad K = HW_K, \quad V = HW_V,$$
$$A = \frac{QK^\text{T}}{\sqrt{d_e}} + B, \quad \text{Attn}(H) = \text{softmax}(A)V,$$

where $W_Q, W_K, W_V \in \mathbb{R}^{d_e \times d_e}$ are the projection matrices, and $B$ is the attention bias given in equation 1. The extension to the multi-head attention is standard and straightforward.

**Further Implementation Details**    The graph Transformer in the experiments contains 9 layers with embedding dimension $d_e = 512$ and 32 self-attention heads. The hidden layer of the FFN module has a width equal to $d_e$. Moreover, we do not include the special node `[VNode]` in the original Graphormer to simplify implementation.

## C.2    INR ARCHITECTURE

In the realm of Implicit Neural Representation (INR), data samples are interpreted as coordinate-based functions, where each function accepts a coordinate $(t, x)$ as input and yields an approximated function value $\hat{u}(t, x)$ at that specific coordinate point. Various architectures of such INRs have been proposed in the literature, including DeepONet (Lu et al., 2021), HyperDeepONet (Lee et al., 2023) for neural operators, as well as SIREN (Sitzmann et al., 2020), WIRE (Saragadam et al., 2023), MFN (Fathony et al., 2021), Poly-INR (Singh et al., 2023) and others (Ramasinghe & Lucey, 2022; Chen & Wang, 2022; Jun & Nichol, 2023) in computer vision. In the experiments, we utilize an adapted version of Poly-INR (Singh et al., 2023), which exhibits better prediction accuracy and training stability compared with other candidates in our setting. Inspired by COIN++ (Dupont et al., 2022), we also employ $L$ hypernets, in which the $\ell$-th hypernet takes $\mu^\ell \in \mathbb{R}^{d_e}$ as its input, and generates the scale- and shift-modulations for the $\ell$-th hidden layer of our Poly-INR.

The intricate architecture of our INR decoder is illustrated in Figure 6, with the mathematical framework detailed below. We take $h_0 = \mathbf{1}$ to be the vector with all entries equal to one. For $\ell = 1, 2, \ldots, L$, we compute

$$g_\ell = W_\ell^\text{in} \begin{bmatrix} t \\ x \end{bmatrix} + b_\ell^\text{in}, \quad s_\ell^\text{scale} = \text{MLP}_\ell^\text{scale}(\mu^\ell), \quad s_\ell^\text{shift} = \text{MLP}_\ell^\text{shift}(\mu^\ell),$$
$$q_\ell = s_\ell^\text{scale} \odot \left( W_\ell^\text{h} (h_{\ell-1} \odot g_\ell) + b_\ell^\text{h} \right) + s_\ell^\text{shift}, \quad h_\ell = \sigma(q_\ell),$$

and the network output is given as $\hat{u}(t, x) = W^\text{Last} h_L + b^\text{Last}$. Here, the activation function $\sigma(\cdot)$ is a leaky-ReLU operation with a slope of $0.2$ at the negative input range, followed by a clipping operation into the interval $[-256, 256]$ to improve training stability. The hypernets correspond to $\text{MLP}_\ell^\text{scale}$ and $\text{MLP}_\ell^\text{shift}$. Note that in the original Poly-INR, the hypernets are utilized to generate $W_\ell^\text{in}$ and $b_\ell^\text{in}$. Compared with our practice of generating $s_\ell^\text{scale}$ and $s_\ell^\text{shift}$, this method exhibits better accuracy, but deteriorates the training efficiency, and is therefore not adopted in our experiments.

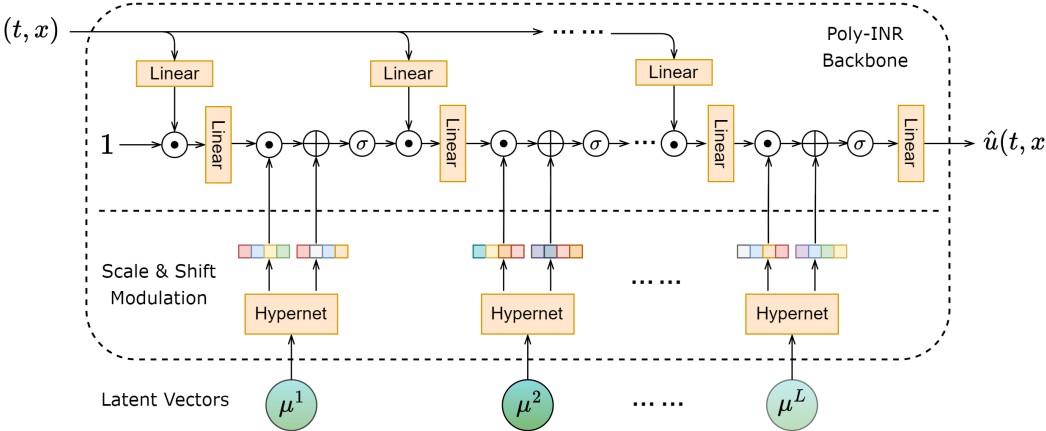

Figure 6: INR decoder architecture of PDEformer.

### C.3 TRAINING SETTING

The experimental settings, including model hyperparameters and configurations, are outlined in Table 2. For a comprehensive understanding of the baseline models employed in our experiments, we provide an overview of all models:

- **DeepONet:** DeepONet employs a unique architecture with two sub-networks: a branch net and a trunk net. The branch net processes a fixed number of sensor observations (256 points from the initial condition in our case), while the trunk net handles coordinate inputs for inference, akin to PDEformer's input mechanism. The outputs from both networks are combined to produce the solution value. Each sub-network consists of a six-layer MLP with 256 hidden neurons and utilizes the ReLU activation function. Notably, DeepONet's mesh-free nature allows for training with scattered data points, enabling us to sample 8192 points per iteration from $256 \times 256$ grids for each data sample during both DeepONet's training and PDEformer's fine-tuning processes.

- **FNO:** The Fourier Neural Operator (FNO) operates on a mesh-dependent yet resolution-independent principle. It initially transforms regular grid data into multi-channel hidden features through a pointwise fully connected layer, followed by processing through several Fourier Layers, and finally map to the solution grid. In Fourier Layer, the FNO keeps the lowest 12 Fourier modes. In our experiments, the FNO2D model is utilized, with the initial condition (256 spatial points) extended to form a $256 \times 256$ input grid, allowing for simultaneous full field output.

- **U-Net:** U-Net adopts a CNN-based encoder-decoder framework, distinguished by its 4 layers of downsampling and upsampling convolutions, bridged by intermediate residual connections. Analogous to FNO2D, both the input and output dimensions are set to $256 \times 256$. Unlike the mesh-free DeepONet or PDEformer, FNO and U-Net require training data organized in regular grids.

- **PDEformer:** The Transformer-based Graphormer is configured with 9 layers, a 512-dimensional embedding space, and 32 attention heads. The Poly-INR part employs $L = 8$ hidden layers with 256 neurons, and each hidden layer is dynamically modulated using separate scale and shift hypernets, each comprising of a 3-layer MLP with independent parameters.

In the pretraining stage of PDEformer, we employ the normalized root-mean-squared-error (nRMSE) loss function due to its effectiveness in improving training efficiency. A learning rate schedule is implemented, progressively reducing the learning rate at predetermined epochs to improve the stability of the training process. Moreover, a warm-up period is utilized at the start of training to mitigate the risk of early training failures by gradually increasing the learning rate from zero to the initial pre-scheduled value.

Table 2: Hyperparameters

| Parameter | Value | Description |
|---|---|---|
| **DeepONet** | | |
| trunk_dim_in | 2 | Input dimension of the trunk network |
| trunk_dim_hidden | 256 | Dimension of hidden features in the trunk network |
| trunk_num_layers | 6 | Number of layers in the trunk network |
| branch_dim_in | 256 | Input dimension of the branch network |
| branch_dim_hidden | 256 | Dimension of hidden features |
| branch_num_layers | 6 | Number of layers in the branch network |
| dim_out | 2048 | Output dimension of the trunk net and the branch net |
| num_tx_samp_pts | 8192 | Number of sample points used per training iteration |
| learning_rate | 0.0003 | The initial learning rate for the optimizer |
| **FNO** | | |
| resolution | 256 | The resolution of the grid |
| modes | 12 | The truncation number of Fourier modes |
| channels | 20 | The number of channels in the hidden layers |
| depths | 4 | The number of Fourier Layers in the neural network |
| learning_rate | 0.0001 | The initial learning rate for the optimizer |
| **U-Net** | | |
| learning_rate | 0.0001 | The initial learning rate for the optimizer |
| **Autoregressive U-Net** | | |
| learning_rate | 0.0001 | The initial learning rate for the optimizer |
| **PDEformer** | | |
| **Graphormer** | | |
| num_patch | 16 | Number of patches used for the initial condition |
| num_layers | 9 | Number of layers in Graphormer |
| embed_dim | 512 | Dimension of the feature embedding |
| ffn_embed_dim | 512 | Dimension of the feed-forward network embedding |
| num_heads | 32 | Number of attention heads |
| pre_layernorm | True | Whether to use layer normalization before each block |
| **Poly-INR** | | |
| dim_in | 2 | Input dimension |
| dim_hidden | 256 | Dimension of the hidden feature |
| dim_out | 1 | Output dimension |
| num_layers | 8 | Number of hidden layers |
| **Layerwise Hypernet** | | |
| hyper_dim_hidden | 256 | Dimension of hidden layers in a hypernet |
| hyper_num_layers | 3 | Number of layers in a hypernet |
| share_hyper | False | Whether hypernets share parameters across all layers |
| **PDEformer Pretraining** | | |
| batch_size | 80 | Total batchsize used in one iteration |
| learning_rate | 0.0003 | The initial learning rate for the optimizer |
| epochs | 1000 | The total number of training epochs |
| loss_type | nRMSE | Use the normalized root-mean-squared-error for training |
| optimizer | Adam | The optimization algorithm |
| lr_scheduler | mstep | The learning rate scheduler |
| lr_milestones | [0.4, 0.6, 0.8] | Epoch milestones for learning rate adjustment |
| lr_decay | 0.5 | Decay factor for reducing the learning rate |
| warmup_epochs | 10 | Epochs to linearly increase the learning rate |

## D METRIC AND DETAILED RESULTS

Throughout this study, we quantify performance using the relative $L^2$ error as our primary metric for testing. The relative $L^2$ error is mathematically represented by the loss function:

$$\mathcal{L}_{\text{relative}} = \frac{\|u - \hat{u}\|_{L^2}}{\|u\|_{L^2}}, \tag{2}$$

where $\|u - \hat{u}\|_{L^2}$ is the $L^2$-distance between the predicted solution $\hat{u}$ and the ground-truth solution $u$, and $\|u\|_{L^2}$ is the $L^2$-norm of the true solution. This metric offers a normalized measure of the error, thereby enabling consistent comparisons across datasets with varying scales and magnitudes.

All the experiments are conducted using MindSpore[5] 2.0, and the pretraining involving $1,000$ epochs takes about 79 hours on 8 NPUs (84 hours if the internal testing evaluations is taken into account). Figure 7 illustrates the pretraining process of PDEformer.

In our investigation of the forward problem, Table 3 compares the prediction results on PDEs with coefficients lying outside the range of the pretraining data. Note that all baseline methods as well as `PDEformer-FS` do not involve a pretraining process. We also embarked on a detailed investigation to assess the model's learning efficiency with limited data. Specifically, we reduced the training dataset size from 9k to 100 and 1k samples. As depicted in Figure 8, the fine-tuned PDEformer model notably excels, outperforming all other methods in the test. Moreover, the zero-shot PDEformer establishes a commendably high benchmark, demonstrating robust performance without any fine-tuning. It is particularly noteworthy that under OoD conditions, such as in the Advection ($\beta = 1$) and Reaction-Diffusion scenarios, the fine-tuned PDEformer rapidly attains superior results. This highlights the model's few-shot learning[6] ability in adapting to unfamiliar scenarios.

In terms of the inverse problem, the additive noise value at each grid point is randomly sampled from $U([-r\|u\|_{L^\infty}, r\|u\|_{L^\infty}])$, where $u$ is the true solution without noise, and $r$ is the noise level. Table 4 shows the recovered coefficients for three PDEs out of the 40 random samples, with the corresponding noisy observations and PDEformer predictions illustrated in Figure 9. Note that the input of PDEformer is the recovered PDE coefficients rather than the ground-truth values. The results implies the promising accuracy of PDEformer in both forward and inverse problems, even in the case of noisy observations.

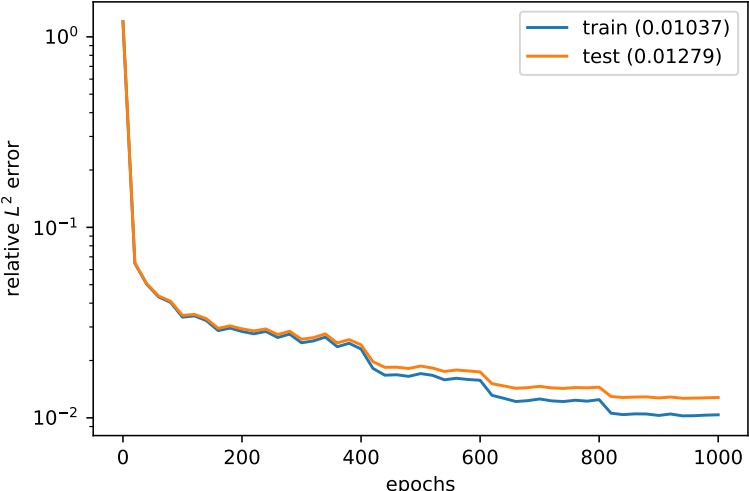

Figure 7: The pretraining process of PDEformer. The learning rate is attenuated by half when the pretraining progress reaches $40\%, 60\%$ and $80\%$. Final train and test loss values are displayed in the legend.

---

[5]https://www.mindspore.cn

[6]In this work, we use the term *few-shot learning* to describe the model's proficiency in adapting to and learning from new data that falls outside the distribution of the training set, using only a small number of examples for fine-tuning.

Table 3: Test relative $L^2$ error on PDEBench, in which the PDE coefficients lie outside the range of the pretraining data. We format the first and second best outcomes in bold and underline, respectively.

| Model | Burgers $\nu = 0.001$ | Advection $\beta = 1$ | Reaction-Diffusion $\nu = 1, \rho = 1$ |
|---|---|---|---|
| U-Net (Ronneberger et al., 2015) | 0.2431 | 0.2655 | 0.0126 |
| Autoregressive U-Net | 0.2865 | 0.3735 | 0.0055 |
| DeepONet (Lu et al., 2021) | 0.2010 | 0.0187 | 0.0015 |
| FNO (Li et al., 2021) | 0.0700 | 0.0097 | 0.0018 |
| PDEformer-FS (Ours) | 0.0645 | 0.0239 | 0.0013 |
| PDEformer (Ours) | 0.0921 | 0.4000 | 0.7399 |
| PDEformer-FT (Ours) | **0.0295** | **0.0075** | **0.0009** |

Figure 8: Variation of test error with number of fine-tuned samples. "PDEformer" represents our model's direct inference capability without the need for fine-tuning. This unique characteristic is visually depicted as a horizontal dashed line across the figure.

## E INFERENCE TIME

Table 5 showcases a comparison of the number of parameters, per-sample inference time and prediction accuracy for a range of models, including DeepONet, FNO, U-Net, and PDEformer. We also include the results of two traditional numerical solvers. The former is based on the first-order upwind finite-difference (FD) scheme, utilizing the `solve_ivp` function provided by the SciPy Python package, and the latter being Dedalus, the spectral-method-based solver employed in generating our ground-truth solution data. The evaluation was conducted using the 1D Advection equation ($\beta = 1.0$) on a $256 \times 256$ spatial-temporal grid as a test case, with neural network mod-

Table 4: Recovered coefficients under different noise levels $r$, in which three PDEs out of the 40 random samples are selected for illustration. The corresponding viscosity coefficients are $\nu = 0.0873, 0.0771$ and $0.0144$ respectively, and do not require recovery.

| PDE form | $r$ | 0 | 0.001 | 0.01 | 0.1 | Reference |
|---|---|---|---|---|---|---|
| $u_t + c_{01}u - \nu u_{xx} = 0$ | $c_{01}$ | 0.0801 | 0.0801 | 0.0801 | 0.0801 | 0.0827 |
| $u_t + (c_{11}u + c_{12}u^2)_x - \nu u_{xx} = 0$ | $c_{11}$ | 1.7260 | 1.7260 | 1.7253 | 1.7090 | 1.7306 |
| | $c_{12}$ | 1.3386 | 1.3386 | 1.3392 | 1.3810 | 1.3398 |
| $u_t + c_{00} + c_{03}u^3 - \nu u_{xx}$ | $c_{00}$ | $-1.0147$ | $-1.0147$ | $-1.0147$ | $-1.0171$ | $-0.9946$ |
| | $c_{03}$ | $-1.1130$ | $-1.1130$ | $-1.1130$ | $-1.1239$ | $-1.1573$ |
| $+(c_{11}u + c_{12}u^2)_x = 0$ | $c_{11}$ | 0.2198 | 0.2198 | 0.2198 | 0.2269 | 0.2045 |
| | $c_{12}$ | 1.0818 | 1.0818 | 1.0818 | 1.0825 | 1.0896 |

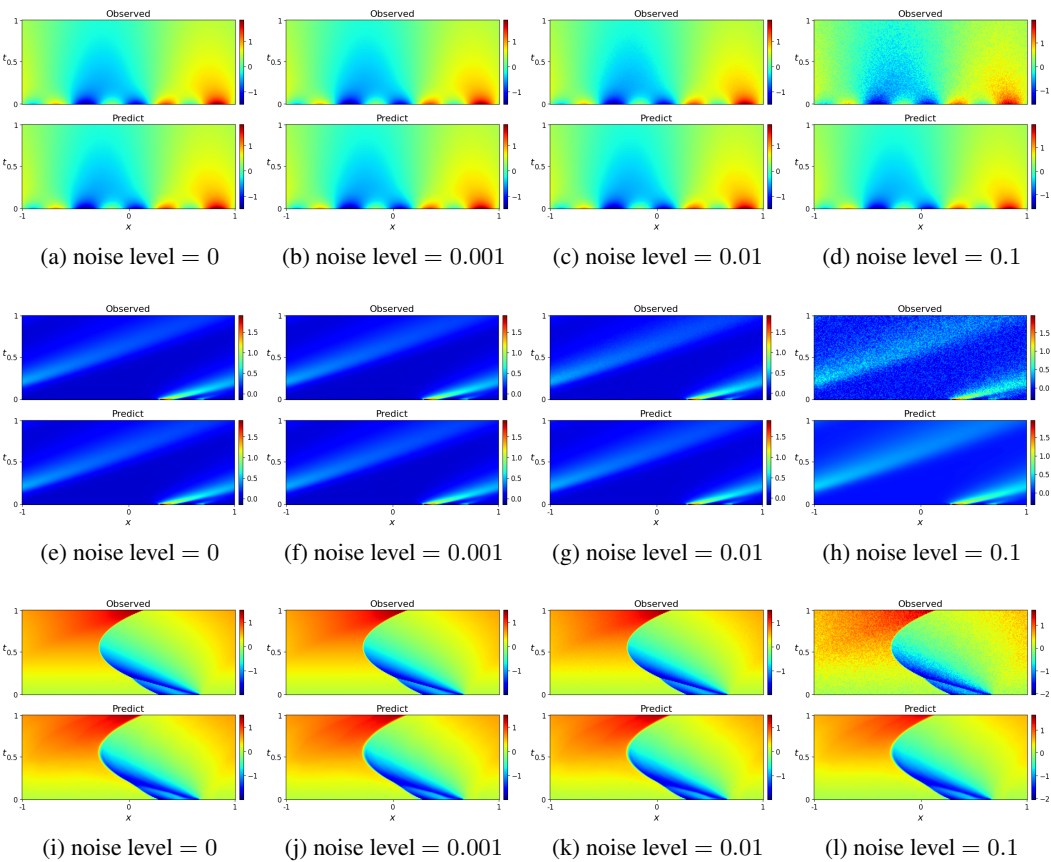

Figure 9: Comparison of noisy observations with predicted solutions employing coefficients derived from inversion as input for PDEformer across diverse noise levels. The three rows correspond to the three equations shown in Table 4.

els tested on a single NPU and traditional solvers executed on a CPU. The neural network models are adequately trained on the corresponding dataset, and the batch size is standardized to 10 during the test. We average the total time consumption of each model across all samples to show the per-sample inference time. As the FD solver exhibits lower accuracy, the spatial grid resolution is refined to $16 \times 256 = 4096$ in its solution process.

Table 5: Comparison of model trainable parameters and per-sample inference time. The relative $L^2$ error of the models has already been presented in Table 1.

| Model | DeepONet | FNO | U-Net | PDEformer | FD | Dedalus |
|---|---|---|---|---|---|---|
| Num. Param. | 1.65M | 0.92M | 13.39M | 19.21M | - | - |
| Infer. Time (ms) | 8.06 | 3.61 | 5.51 | 8.76 | 2072.3 | 410.8 |
| Rel. $L^2$ Error | 0.0187 | 0.0097 | 0.2655 | 0.0075 | 0.0674 | - |

While the comparison reveals a significantly longer inference time for Dedalus, it's essential to acknowledge the inherent differences in the computational platforms and the nature of the models themselves. This juxtaposition, though not strictly fair, aims to illustrate the potential efficiency of machine learning methods in solving PDEs.

## F  AUTOREGRESSIVE U-NET

The U-Net model exhibits unsatisfactory performance in our experiments, and some may speculate that the practice of predicting the entire spatial-temporal solution is not suitable for U-Nets. To address these concerns, we also implement an autoregressive variant of the U-Net model. Following PDEBench Takamoto et al. (2022), this model takes $\ell$ consecutive timesteps as the input, and predicts the next unknown timestep. In other words, the model approximates the mapping $[u(t-\ell, \cdot), \ldots, u(t-1, \cdot)] \mapsto \hat{u}(t, \cdot)$. The model architecture is analogous to the non-autoregressive U-Net, except that it now operates on one-dimensional data.

During training, we randomly select $\ell$ consecutive timesteps from a data sample, feed it into the U-Net model, and rollout to predict the next $K$ timesteps:

$$[u(t-\ell, \cdot), \ldots, u(t-2, \cdot), u(t-1, \cdot)] \mapsto \hat{u}(t, \cdot),$$
$$[u(t-\ell+1, \cdot), \ldots, u(t-1, \cdot), \hat{u}(t, \cdot)] \mapsto \hat{u}(t+1, \cdot),$$
$$\cdots$$
$$[\hat{u}(t-\ell+K-1, \cdot), \ldots, \hat{u}(t+K-2, \cdot)] \mapsto \hat{u}(t+K-1, \cdot).$$

The loss function is a weighted average of the prediction error, in the form

$$\mathcal{L}(\theta) = \sum_{k=0}^{K-1} \lambda_k \cdot \mathrm{nRMSE}(u(t+k, \cdot), \hat{u}(t+k, \cdot)).$$

In the implementation, we select $\ell = 4$, $K = 16$, $\lambda_0 = 1$, $\lambda_1 = \cdots = \lambda_{15} = 0.1$.

For the inference phase, we feed the first $\ell$ timesteps into the model, and rollout until we obtain the entire spatial-temporal solution. Note that all the other models involved in our experiments only takes the initial value (i.e. the first timestep) as the network input. Figure 10 illustrates the predictions of the autoregressive U-Net model. We notice that the model successfully captures the overall dynamics inside the spatial interval, and exhibits a high per-step prediction accuracy. However, small error would appear near the boundary points, and is then amplified during the rollout prediction process, leading to unsatisfactory spatial-temporal prediction results. Indeed, such boundary errors might be mitigated if we modify the network architecture to enforce periodicity, but the resulting network design would then be equation-specific, and is not applicable to more general PDEs with non-periodic boundary conditions.

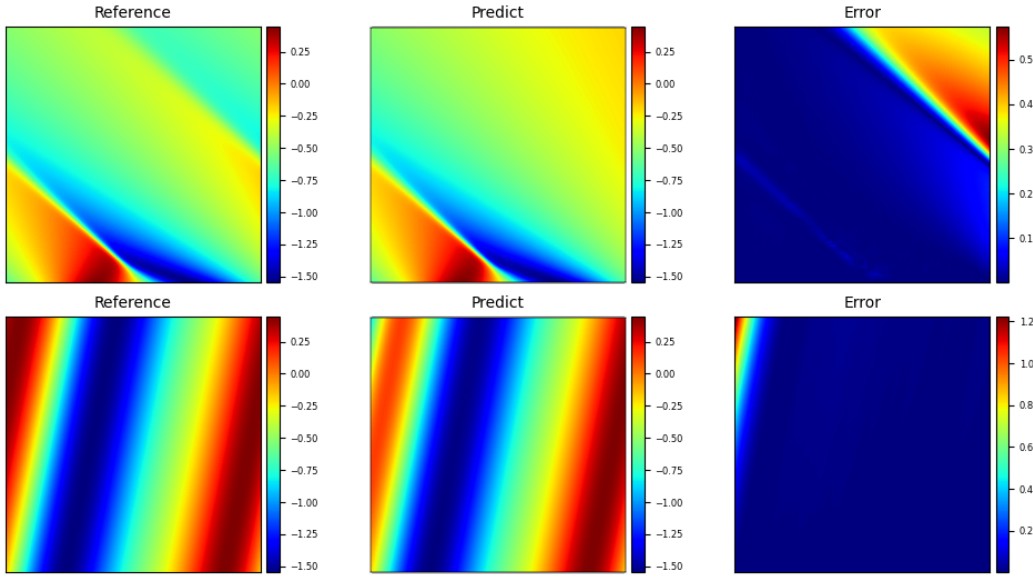

Figure 10: Prediction results of the autoregressive U-Net model. Top: Burgers' equation with $\nu = 0.1$. Bottom: Advection equation with $\beta = 0.1$. The horizontal axis corresponds to the spatial coordinate $x$, and the vertical axis corresponds to the temporal axis $t$.

