# OpenReview forum: "PDEformer: Towards a Foundation Model for One-Dimensional Partial Differential Equations"
_ICLR.cc/2024/Workshop/AI4DiffEqtnsInSci — AI4DiffEqtnsInSci @ ICLR 2024 Poster_

### Official Review · Reviewer_V3Fd · 2024-02-27
**Good paper**

**Rating:** 8
**Confidence:** 4

**Review:**

The paper is in the direction of creating foundation models for PDEs. I have no comment as the paper is nicely written and has great results.

---

### Official Review · Reviewer_6gPK · 2024-02-27
**Torn between recommending acceptance and rejection; interesting architecture but multiple concerns**

**Rating:** 5
**Confidence:** 4

**Review:**

**Explanation of paper**

This paper introduces a method for solving 1D time-dependent PDEs which they call PDEformer. The unique (or at least unusual) feature of the PDEformer architecture is that it represents the PDE being solved via a computational graph (see graph construction, figure 1). The symbolic form of the PDE is thus an input to PDEformer, and by doing by PDEformer is able to solve multiple different classes of PDEs. The goal of the paper is to develop a “noteworthy milestone” towards a so-called foundation model for PDEs, which can *generalize* over many different types of PDEs. This is certainly a worthy goal and of interest to the workshop.

I have three primary concerns about this paper.

**Result in abstract is inconsistent with results in section 3**

The abstract states the following:
> Following pretraining on data exhibiting a certain level of diversity, our model achieves zero-shot accuracies on benchmark datasets that surpass those of adequately trained expert models.

Table 1 compares five models across six tests. The three expert models are U-Net, DeepONet, and FNO. These are compared to the zero-shot PDEformer and the fine-tuned PDEformer. Table 1 shows that, on four of the six tests, the zero-shot PDEformer model does worse than the expert models. It is incorrect to say that the model does better than expert models when it does worse on the majority of benchmark problems. The abstract of the paper should instead say

> Following pretraining on data exhibiting a certain level of diversity, our model performs worse than adequately trained expert models on most benchmark datasets.

The mistake in this case was that the result reported in the abstract "[omitted] outcomes which are deemed to be unfavourable or statistically insignificant." This is called [outcome reporting bias](https://catalogofbias.org/biases/outcome-reporting-bias/), a form of [reporting bias](https://catalogofbias.org/biases/reporting-biases/). Reporting biases are a form of scientific misconduct.


**Weak U-Net baseline in table 1**

Table 1 compares five models across six tests. A few days ago, I spent a couple hours going through the literature, looking at other papers that reported the relative L2 error (sometimes called the “percent error”). I don’t have time to go back through the papers that I found and cite them all, but I concluded that the FNO results in table 1 all seemed reasonable and consistent with the relative L2 error reported in previous papers. I think the FNO baseline is implemented correctly and is a strong baseline.

My concern about a possible weak baseline is with the U-Net. I couldn’t find U-Nets that had solved these benchmark PDEs, but I know that on other 1D and 2D benchmark PDEs (such as the Kuramoto-Sivashinsky equation, the incompressible Euler equation, etc) U-Nets were found to work better than FNOs at solving PDEs [2,3,4]. Thus, the terrible performance of U-Nets on every benchmark problem is surprising and suggests that the U-Nets were not implemented correctly and are a weak baseline.

I believe the reason that U-Nets achieved such unexpectedly poor performance was that they were not implemented as a 1D autoregressive model (see, for example, [4]) but as a full-solution model that outputs the entire 2D (space & time) solution all at once. For this reason, the U-Net baseline is almost certainly a weak baseline.

The input should have been a 1D 256-length array, and the output the solution or change in solution at the next timestep. Instead, the input was a 256x256 array, and the output was the solution over every timestep.

Given the results of [2,3,4] where autoregressive U-Nets outperform FNOs on 1D and 2D PDEs, I would expect the same would likely occur in table 1. It is quite possible that autoregressive U-Nets would outperform PDEformer on this problem.

**Failure to identify the sources of empirical gains**

Please read *Troubling trends in machine learning scholarship* [5].  [5] discusses four troubling trends, the first two of which are

> 1. Failure to distinguish between explanation and speculation.
> 2. Failure to identify the sources of empirical gains, e.g. emphasizing unnecessary modifications to neural architectures when gains actually stem from hyper-parameter tuning.

This paper makes two statements which purport to explain, but actually are speculating. I believe that both statements are incorrectly identifying the source of empirical gains, or at minimum failing to provide sufficient evidence for the source of empirical gains.


Statement #1:
> This superior performance stems from the pretrained PDEformer’s exposure to diverse PDEs during pretraining, from which the model learns a generalizable law to outperform models trained specifically for individual PDEs.

The paper fails to provide enough evidence to make this conclusion. PDEformer trains on 500k samples of diverse PDEs, then (in 2 out of 6 instances) outperforms expert models trained on 9k samples. Yet it is very much possible that the training on diverse PDEs is not the reason that PDEformer does better in those 2 instances. Perhaps the paper is emphasizing unnecessary modifications to training procedures when gains actually stem from an improved architecture.

To test whether it is the training procedure or the architecture that causes these empirical gains, the PDEformer architecture trained only on the 9k samples should be compared to the other expert models trained only on the 9k samples. Performing such an ablation study would allow us to identify the source of empirical gains.

Statement #2:
> This contrast highlights PDEformer’s swift and accurate adaptability, marking a significant advancement over neural operators.

Again, the paper does not provide enough evidence to conclude that PDEformers are more adaptable than neural operators. If neural operators were trained on 500k samples with diverse PDEs, perhaps they too would be more capable of being trained on a small dataset with fewer iterations. It is possible that the neural network initializations, not something inherently better about PDEformer than neural operators, is the reason for the improved adaptability. Perhaps the paper is emphasizing unnecessary modifications to architectures when gains actually stem from improved initialization.

To test whether it is the architecture or the initialization that causes these empirical gains, instead of training an FNO from scratch, compare fine-tuning of a PDEformer to retraining an FNO originally trained for a different task.


**Additional comments:**
* Line 2: is “advocate” really the right word to use here? I don’t think so.
* "By leveraging the power of generality …” What is the power of generality? Can you explain what it is and provide references?
* What does “adequately trained” mean? What does “sufficiently trained” mean? In figure 3, the train and test loss for the FNO are both still decreasing on a logarithmic scale. Shouldn’t they be trained longer?
* How do I know that the “patch nodes” are helpful? Are they? Can you perform an ablation study where you test with and without these patch nodes?


**Conclusion:**

I am torn between recommending acceptance and recommending rejection. On the one hand, this paper introduces an interesting architecture that is likely of interest to the workshop. On the other hand, I have concerns about reporting bias in the abstract (a form of scientific misconduct), a weak baseline (potentially leading to inaccurate conclusions), and a set of conclusions which aren’t supported by the experiments performed in the paper.

My personal recommendation would be to reject this paper. While reporting bias is sadly quite common in ML research, just because scientific misconduct is common does not mean it should be tolerated. When papers attempt to blatantly mislead readers — doing worse than baselines, but claiming to do better — I think that is clear grounds for rejection.

I will ultimately leave it up to the editor to make the final decision.


[1] Zhuang, Jiawei, et al. "Learned discretizations for passive scalar advection in a two-dimensional turbulent flow." Physical Review Fluids 6.6 (2021): 064605.

[2] Stachenfeld, Kimberly, et al. "Learned coarse models for efficient turbulence simulation." arXiv preprint arXiv:2112.15275 (2021).

[3] Gupta, Jayesh K., and Johannes Brandstetter. "Towards multi-spatiotemporal-scale generalized pde modeling." arXiv preprint arXiv:2209.15616 (2022).

[4] Lippe, Phillip, et al. "Pde-refiner: Achieving accurate long rollouts with neural pde solvers." Advances in Neural Information Processing Systems 36 (2024).

[5] Lipton, Zachary C., and Jacob Steinhardt. "Troubling trends in machine learning scholarship." arXiv preprint arXiv:1807.03341 (2018).

---

### Meta-Review · Area_Chair_qezv · 2024-02-27

**Recommendation:** Accept (Poster)

**Metareview:**

This paper is relevant to the paper since it investigates developing a Transformer-based foundation model for PDEs. In particular, it tests how to generalize across various PDEs, which is very important. This was a borderline paper. I am voting for acceptance but in the camera ready paper the authors should address the valid points from Reviewer 6gPK by correcting the abstract and double checking their U-Net baseline.

---

### Decision · Program_Chairs · 2024-02-28

Accept (Poster)